# Crotoxin Modulates Macrophage Phenotypic Reprogramming

**DOI:** 10.3390/toxins15100616

**Published:** 2023-10-17

**Authors:** Camila Lima Neves, Christiano Marcello Vaz Barbosa, Priscila Andrade Ranéia-Silva, Eliana L. Faquim-Mauro, Sandra Coccuzzo Sampaio

**Affiliations:** 1Laboratory of Pathophysiology, Butantan Institute, São Paulo 05503-900, Brazil; camila.neves@butantan.gov.br; 2Department of Biochemistry, Federal University of São Paulo, São Paulo 04044-020, Brazil; cmvbarbosa@unifesp.br; 3Department of Clinical Medicine, University of Ribeirão Preto, Campus Guarujá, São Paulo 11440-003, Brazil; parsilva@unaerp.br; 4Laboratory of Immunopathology, Butantan Institute, São Paulo 05503-900, Brazil; eliana.faquim@butantan.gov.br; 5Department of Immunology, Institute of Biomedical Sciences, University of São Paulo, São Paulo 05508-220, Brazil; 6Department of Pharmacology, Institute of Biomedical Sciences, University of São Paulo, São Paulo 05508-220, Brazil

**Keywords:** rattlesnake, macrophage plasticity, cytokines, tumor microenvironment, immunomodulatory effect

## Abstract

Macrophage plasticity is a fundamental feature of the immune response since it favors the rapid and adequate change of the functional phenotype in response to the pathogen or the microenvironment. Several studies have shown that Crotoxin (CTX), the major toxin of the *Crotalus durissus terrificus* snake venom, has a long-lasting antitumor effect both in experimental models and in clinical trials. In this study, we show the CTX effect on the phenotypic reprogramming of macrophages in the mesenchymal tumor microenvironment or those obtained from the peritoneal cavity of healthy animals. CTX (0.9 or 5 μg/animal subcutaneously) administered concomitantly with intraperitoneal inoculation of tumor cells (1 × 10^7^/0.5 mL, injected intraperitoneally) of Ehrlich Ascitic Tumor (EAT) modulated the macrophages phenotype (M1), accompanied by increased NO^•^ production by cells from ascites, and was evaluated after 13 days. On the other hand, in healthy animals, the phenotypic profile of macrophages was modulated in a dose-dependent way at 0.9 μg/animal: M1 and at 5.0 μg/animal: M2; this was accompanied by increased NO^•^ production by peritoneal macrophages only for the dose of 0.9 μg/animal of CTX. This study shows that a single administration of CTX interferes with the phenotypic reprogramming of macrophages, as well as with the secretory state of cells from ascites, influencing events involved with mesenchymal tumor progression. These findings may favor the selection of new therapeutic targets to correct compromised immunity in different systems.

## 1. Introduction

At the tumor site, macrophages, called tumor-associated macrophages (TAM), influence fundamental aspects of tumor biology: they produce molecules that directly affect the growth, motility and invasion of tumor cells, intensify neoangiogenesis, regulate inflammatory and adaptive immune responses and catalyze important structural changes in the extracellular matrix (ECM) [1,2,3]. Regarding the ECM, its constituent proteins are continuously produced and degraded by many cell types, such as tumor, endothelial and stromal cells, with TAMs being the major regulators, leading to the marked presence of matrix components that are not normally found [2]. Therefore, TAMs present characteristics very close to those presented by macrophages with the M2 phenotype [4]. Additionally, Sica and colleagues [5] demonstrated that TAMs can have a dual function since they are capable of secreting pro- and anti-inflammatory mediators, which can compromise the immune response and tumor development.

Experimentally, the migration and infiltration of monocytes to the tumor mass seem to be specifically regulated by factors originating from the tumor, such as chemoattractant substances [6] and extracellular matrix proteins, which can attract and activate macrophages directly [7]. These, in turn, produce tumor necrosis factor-alpha (TNF-α) and nitric oxide (NO^•^) that amplify the migration of macrophages to the tumor site [7]. Peritoneal cells obtained from tumor-bearing animals (fibrosarcoma) show high production of superoxide and increased production of NO^•^, in the phase of tumor rejection [7,8]. During the development of ascitic tumors, a large amount of M2 phenotype macrophages is found, leading to an important suppression of the immune system and increased proliferation of tumor cells [9]. Subsequently, it was demonstrated that monocytes present in the bone marrow of tumor-bearing animals are already polarized into M1 and M2, according to the stage of tumor development [10]. These characteristics associate the M2 phenotype with the development of tumors; therefore, the reprogramming of tumor-associated macrophages (from M2 to M1) is considered an important immunotherapeutic possibility for the control of tumor progression [11,12,13].

Crotoxin (CTX) was isolated by Slotta and Fraenkel-Conrat, in 1938 [14], and its structure was described by Fraenkel-Conrat and Singer (1956) [15] as being a heterodimeric β-neurotoxin formed by non-covalent association of two different subunits: crotapotin (CA) and phospholipase A_2_ (PLA_2_—CB) [16,17,18,19]. Furthermore, it is possible to find sixteen different isoforms of CTX in the venom of *Crotalus durissus terrificus*, with there being four isoforms of the CA subunit (CA_1_, CA_2_, CA_3_ and CA_4_) and four isoforms of the CB subunit (CBa2, CBb, CBc and CBd). These isoforms of CA and CB can be combined randomly, giving rise to complexes with distinct pharmacological and biological properties [20]. The antitumor potential of CTX has been well demonstrated in different experimental studies in vivo and in vitro [19], as well as in clinical studies [21,22]. Studies carried out in our laboratory have demonstrated the importance of the long-term immunomodulatory activity of CTX on macrophages for tumor progression [23].

In order to expand the mechanisms involved in the antitumor effect of CTX, the objective of this study was to evaluate the action of this toxin on the phenotypic profile of macrophages. For this, resident macrophages and macrophage reprogramming induced by the tumor microenvironment were evaluated after a long period of a single administration of CTX and on macrophage reprogramming induced by the tumor microenvironment after a long period of a single administration of CTX.

## 2. Results

### 2.1. Developmental Characterization of Ehrlich Ascitic Tumor—EAT

During the phases of EAT development (lag phase: 1–5 days after cell inoculation, log phase: 6–10 days and terminal period: 11–15 days), intraperitoneal injection of 1 × 10^7^ cells in 0.5 mL resulted in the exponential development observed by the total number of cells (Figure 1A) present in the ascitic fluid. This total number of cells corresponds to tumor cells and leukocytes. Figure 1B shows that the volume of ascitic fluid increases progressively and significantly after the 6th day of EAT inoculation.

### 2.2. Effect of CTX on Ascitic Fluid Volume

A single administration of CTX, of both doses (0.9 µg/animal or 5.0 µg/animal), significantly reduces (39% and 33%, respectively) the volume of ascites, on the 6th day of inoculation of EAT cells (lag phase for the log phase), when compared to the saline-treated control group (Figure 2).

When the action of CTX was evaluated from the log phase to the terminal phase *(*13th day), it was observed that only the dose of 0.9 µg/animal of CTX was able to significantly reduce (27%) the volume of ascites, when compared to the control group consisting of animals treated with saline under the same experimental conditions (Figure 2).

### 2.3. Effect of CTX on Cellularity Present in the Ascitic Fluid

#### 2.3.1. Effect of Concomitant Treatment (6th Day—Lag to Log Phase)

In animals with tumors, the treatment with CTX doses reduced (12% and 18% for 0.9 and 5.0 µg/animal, respectively) the total number of cells (leukocytes and tumor cells) present in the ascitic fluid, after 6 days of EAT inoculation, when compared to the control group (EAT + saline) (Figure 3A). Animals treated with CTX doses showed a reduction (0.9 µg/animal: 16% and 5.0 µg/animal: 19%) of the number of tumor cells in the period evaluated (after 6 days), as shown in Figure 3B. In Figure 3C, the treatment with CTX at a dose of 0.9 µg/animal increased (35%) the number of leukocytes after 6 days of EAT inoculation, when compared to the control group (EAT + saline). Also, among treatments with CTX, the dose of 5.0 µg/animal reduced (29%) the number of leukocytes, after 6 days of EAT inoculation, when compared to the treated group (EAT + CTX 0.9 µg/animal). No significant difference was observed between the tumor-bearing animals treated with CTX at a dose of 5.0 µg/animal and the control animals treated with saline (Figure 3C).

In parallel, healthy animals were treated with the same doses of CTX to evaluate the per se action of the toxin on the number of total leukocytes present in the peritoneal cavity of mice. Animals treated with CTX doses, on the 6th day, exhibited an increase in the number of leukocytes (0.9 µg/animal: 13% and 5.0 µg/animal: 17%) when compared to the control group, which were only injected with saline (s.c.) (Figure 3C).

#### 2.3.2. Effect of Concomitant Treatment (13th Day—Log to Terminal Phase)

In the animals with tumors, both doses of CTX reduced, slightly, without statistical significance, the total number of cells present in the ascitic fluid (0.9 µg/animal: 13%; 5.0 µg/animal: 10%), when compared to the control group (EAT + saline) (Figure 3D). Furthermore, the treatment with CTX did not significantly alter the number of cells (leukocytes or tumors), when compared to the control group (EAT + saline) (Figure 3E). As shown in Figure 3F, the healthy animals were treated with the same doses of CTX, subcutaneously, to evaluate the per se action of the toxin on the total number of leukocytes present in the peritoneal cavity of mice. The results demonstrate that the animals treated with CTX doses, on the 13th day, showed a number of leukocytes like the number found in the control group, which were injected with saline only (s.c.) (Figure 3F).

### 2.4. Effect CTX on Cytokine Release

As shown in Table 1, on the 6th day of the EAT administration, the CTX-dose of 0.9 and 5.0 µg/animal significant reduction of IL-10, below the threshold of detection of the kits used to determine this cytokine (undetectable-N.D.) (Table 1), when compared to the quantification obtained in the EAT-group control (EAT + saline). The secretion of IL1-β, or TNF-α release in the ascitic fluid was not altered by the different doses of CTX when compared to the values obtained from the control animals (EAT + saline). In parallel, the administration of different doses of CTX in animals without tumors did not change the secretion or release of cytokines, when compared to control animals (PBS + saline), as shown in Table 1.

When the profile of cytokine secretion and release was evaluated on the 13th day of EAT inoculation (Table 2), no significant toxin-induced changes were observed. Likewise, in tumor-free mice, the different doses of CTX also did not change the secretory profile, when compared to animals treated with saline (Table 2).

### 2.5. Effect of CTX on Nitric Oxide Production (NO^•^)

The animals with EAT and treated with CTX doses (0.9 and 5.0 µg/animal), on the 6th day of inoculation of tumor cells, showed no changes in the production of secreted NO^•^, when compared to the control group (EAT + saline) (Figure 4). In parallel, the administration of CTX doses of 0.9 and 5.0 µg/animal in non-tumor-bearing animals did not alter the production of NO^•^ secreted by leukocytes in the peritoneal cavity of these animals over 6 days after a single administration of the CTX, when compared to the control group (PBS + saline), as shown in Figure 4.

On the 13th day of the tumor induction and CTX administration of 0.9 μg/animal, a significant increase (35%) in the NO production by macrophages in the supernatant obtained from tumor-bearing animals, when compared to the control group (EAT + saline), was verified. On the other hand, the dose of 5.0 µg/animal of CTX did not change the production of NO^•^ secreted by macrophages in the supernatant of the animals with EAT, when compared to both the control group (EAT + saline) and the group treated with CTX 0.9 µg/animal. With regard to the non-tumor-bearing animals, the dose of CTX 0.9 µg/animal was able to significantly increase NO^•^ production (17%) by macrophages when compared to both the control (PBS + saline) group and the group treated with the dose of 5.0 µg/animal of CTX. In this experimental condition, the dose of CTX 5.0 µg/animal did not change the production of NO^•^ by macrophages, when compared to control animals (PBS + saline), as shown in Figure 4.

### 2.6. CTX Alters the Phenotypic Profile of Resident Macrophages and the Tumor Microenvironment

After characterizing the modulation of CTX on ascites development (volume) caused by EAT, as well as on cellularity, during the transition from lag to log and from log to terminal phases, the prolonged effect of the toxin on the phenotypic reprogramming of macrophages on the 13th day of EAT inoculation was investigated. Initially, to characterize the prolonged effect of CTX on macrophage phenotypic reprogramming, non-tumor-bearing animals were treated with the toxin at different doses (0.9 µg/animal and 5.0 µg/animal, s.c.) or only saline (control animals), and on the 13th day of a single treatment, macrophages were obtained from the peritoneal cavity for immunotyping.

Figure 5A shows the macrophages obtained from non-tumor-bearing animals treated with a dose of 0.9 µg/animal of CTX, showing no change in the percentage of M1 macrophages (CD45^+^F4/80^+^CD68^+^ cells) in the abdominal cavity when compared to the control group (PBS + saline). On the other hand, the dose of CTX 5.0 µg/animal significantly reduced the percentage of M1 macrophages, when compared to the control (PBS + saline: 64%) group. Regarding the profile of M2 macrophages (CD45^+^F4/80^+^CD206^+^ cells), Figure 5A shows that animals without tumors treated with a dose of 5.0 µg/animal CTX showed a significant increase in the percentage of M2 macrophages when compared to the control group (PBS + saline: 52%) and the experimental group treated with CTX (0.9 µg/animal: 67%).

The results presented in Figure 5B demonstrate that, on the 13th day of EAT inoculation, the tumor-bearing animals treated with saline (control) showed a significant reduction in the percentage of M1 macrophages (77%), when compared to the non-tumor-bearing control animals (PBS + saline, Figure 5A). When the animals were treated with different doses of CTX concomitantly with the inoculation of EAT, it was observed that the dose of CTX 0.9 µg/animal (Figure 5B) was able to maintain the percentage of M1 macrophages (60%), which was similar to the non-tumor-bearing animals (Figure 5A, PBS + CTX 0.9 µg/animal). Furthermore, it was noted that the dose of CTX 5.0 µg/animal induced a higher percentage of M1, in the same way as the dose of CTX 0.9 µg/animal did, compared to the control group, which received saline subcutaneously (Figure 5B, EAT + Saline), and compared to the group without tumors treated with CTX 5.0 µg/animal (Figure 5A).

Animals inoculated with EAT and treated concomitantly with doses of CTX 0.9 or 5.0 µg/animal showed a significant decrease in the percentage of M2 macrophages (90% and 80%, respectively), compared to the control group (EAT + saline), as demonstrated in Figure 5B. No significant differences were observed in the percentage of M2 profile between CTX doses in both conditions (with or without tumor).

## 3. Discussion

In the present study, a tumor development model was established through the induction of ascitic fluid formation from the inoculation of Ehrlich tumor cells (EAT) in order to evaluate the action of CTX on the phenotypic profile (the phenotypic status of the evaluated macrophages (M1 or M2) of macrophages obtained from tumor ascites.

Initially, it is important to comment on the choice of tested CTX doses. Cardoso and colleagues [24] demonstrated that subcutaneous administration of CTX at a dose of 5.0 µg/animal temporarily decreases the number of monocytes and lymphocytes with an increase in the number of neutrophils in peripheral blood. These changes are accompanied by an increase of IL-10 and IL-6 in the serum of the mice. Nunes and colleagues [25] demonstrated that CTX has an anti-inflammatory effect when administered at a dose of 0.9 µg/animal, s.c., in animals stimulated by the phlogistic agent carrageenan in the peritoneal cavity. This effect is long-lasting, being observed for up to 7 days after a single administration, with the involvement of the lipoxygenase pathway [25]. Based on these facts, we tested whether these doses would lead to distinct modulation on the onset of ascites, as well as on the phenotypic profile of macrophages.

During the temporal evaluation of the EAT, using BALB/c mice, we observed, under our experimental conditions, that the appearance of ascitic fluid occurs between the 3rd and 6th days after inoculation of the tumor cells (in the transition from the lag phase to the log), remaining increasing until the 13th day of inoculation, the period in which tumor ascites is already established, without the animals either having presented behavioral changes suggestive of pain (spontaneous pain, for example) or changes in weight evolution (weight loss). This increase in ascitic volume was accompanied by an increase in the number of tumor cells and leukocytes, progressively, until the 13th day. Therefore, the main limitation of the model used was that it was not possible to monitor the survival of the animals. The increase in the number of peritoneal leukocytes was observed during tumor development in the three evaluated periods. This data can be explained by the fact that, in a tumor condition, the bone marrow increases the production of cells without an increase in the number of cells in the bloodstream, that is, after reaching the circulation, the cells from the bone marrow migrate immediately to the peritoneum to eliminate the tumor [26]. It is important to emphasize that the analyses of the weight evolution, both in animals with EAT and in animals without tumors, after treatments with CTX are presented as SM-1.

Regarding the treatments with CTX, in the 2nd protocol, the animals received different doses of CTX (0.9 or 5.0 µg/animal, subcutaneously) concomitantly with the inoculation of Ehrlich tumor cells. On the 13th day of these procedures, the phenotypic analysis of macrophages obtained from ascites was performed. Our results showed that on the 13th day of the ascites onset, an important decrease in the population of M1 macrophages was observed in the EAT-mice group, unlike that found in the healthy animals, which present 60% of this population. Also, when ascites was installed, the control animals (EAT-mice group treated only with saline) had a higher population of M2, when compared to the PBS groups that received saline. These data show, therefore, that our model was effective in altering the phenotype of the macrophage populations, which was the main objective of the tumor model employed. Regarding the effects of different CTX doses, the data showed that the dose of 0.9 µg/animal was more effective in maintaining the M1 phenotype in macrophages obtained both from the ascitic fluid (EAT-group) and from the peritoneal cavity of the healthy animals. On the other hand, the dose of 5.0 µg/animal induced M2 phenotype in quiescent resident macrophages (obtained from peritoneal cavity lavage of non-tumor-bearing animals) but stimulated the expression of the M1 phenotype in macrophages obtained from ascitic fluid, suggesting that the modulation of this dose can be stimulus-dependent.

Also, in relation to the results obtained from the 2nd treatment protocol (concomitant), CTX, at a dose of 0.9 µg/animal, inhibited the ascitic volume and stimulated the production of NO. Regarding ascitic volume, Sugiura [27] demonstrated that when fresh ascitic fluid, containing 1 million tumor cells in 0.1 mL, is inoculated into the peritoneal cavity of mice, the cells proliferate on the surfaces of the visceral and parietal peritoneum. During the first three days, ascites is not observed, but the increase in the number of cells is evidenced, that is, the cells are in the process of multiplication in this period, corroborating our results of the temporal evaluation. The observed ascitic fluid accumulation involves increased vascular permeability and pronounced neovascularization of the parietal peritoneum accompanied by high levels of VEGF (vascular endothelial growth factor), especially in gastric, colorectal and ovarian tumors and is also found in the EAT model [28,29].

It is important to highlight that, although no change was observed in the amount of tumor cells, a difference was observed in the accumulation of ascitic fluid with a dose of 0.9 µg/animal, suggesting that CTX interferes with the progressive increase of ascitic fluid in the peritoneum. In fact, our group has been demonstrating that CTX has antiangiogenic activity via macrophage secretory activity [30] and, in this case, involves the reduction of VEGF. Therefore, the data obtained with the dose of 0.9 µg/animal reinforce that the modulation of the macrophage phenotype may be important for some elements involved in tumor progression, including angiogenesis itself.

Regarding the 1st treatment protocol (parameters evaluated on the 6th day after the treatment concomitantly with the inoculation of tumor cells), it was observed that the different doses of CTX inhibited the proliferation of tumor cells and intensified the migration of leukocytes to the cavity of the non-bearing tumor animals. However, only the dose of 0.9 µg/animal maintained the number of leukocytes (mononuclear) migrating to the peritoneal cavity in animals with EAT.

CTX administered concomitantly with the inoculation of tumor cells and the peritoneal lavage performed on the 6th day of this procedure no change was observed in the production of NO**^•^** or in the secretion of the IL-1β. Still, regarding the effect of CTX on the secretion of cytokines detected in the ascitic fluid or in the peritoneal lavage, the concomitant administration to the inoculation of tumor cells inhibited the secretion of IL-10 on the 6th day. It is interesting to point out that, as demonstrated above, the concomitant administration of CTX, regardless of the administered dose, inhibited the secretion of IL-10 in the ascitic fluid of animals with EAT, when detected, on the 6th day of induction. This cytokine has an important suppressive activity on the M1 phenotype of macrophages obtained from animals with EAT [31]. Previously, it was demonstrated that the dose of 5.0 µg/animal triggers the plasma production of IL-10 [24], while the dose of 0.9 µg/animal stimulates lipid mediators of the lipoxygenase pathway in the model of carrageenan [25] and in models evaluated in mice. In rats, a significant increase in lipoxin A_4_ (LXA_4_) and its analogue 15-Epi-LXA_4_ was detected in the plasma of animals without and with Walker 256 tumors after subcutaneous administration of CTX [32]. The increase in these lipid mediators was accompanied by a significant reduction in tumor growth, mainly by inhibiting the number and thickness of the neovessels present in the tumor mass. In this sense, a recent study shows the ability of LXA_4_ to reprogram the M2 phenotype to M1, favoring the inhibition of tumor progression [33] and reinforcing the importance of LXA_4_ and its stable analogue induced by CTX in the tumor context. Additionally, LXA_4_ is an important inducer of NO**^•^** production [34,35]. Interestingly, the dose of the 0.9 µg/animal was the only one capable of inducing an increase in NO**^•^** production. Therefore, we can suggest that this dose is capable of increasing LXA_4_ secretion, leading to the production of NO**^•^**, which may contribute to the inhibition of the tumor’s suppressive action on macrophage polarization, thus keeping them with an M1 phenotype. We attribute this increase in NO**^•^** to leukocytes, mostly macrophages (90%), found in ascitic fluid, since in vivo and in vitro CTX cause important production of this mediator by these cells, from 2 to 48 h of incubation in vitro and up to 14 days after in vivo treatment, in the absence or presence of the tumor microenvironment [23].

Thus, we can suggest that the stage of migration of macrophages to the cavity may involve specific stages of activation of this cell, providing a distinct action of CTX on the release of cytokines or even other mediators such as NO**^•^** and LXA_4_/15-Epi-LXA_4_, probably to control some events involved with tumor progression, such as angiogenesis. This hypothesis may, therefore, explain, at least in part, the distinct actions observed in the different treatment protocols obtained in the present study.

Another interesting aspect to be highlighted is that experimental studies and clinical trials show that the antitumor effect of CTX was observed mainly in solid tumors [21]. The ascitic model showed the antitumor activity of CTX, however, of lower magnitude, when compared to solid tumor models developed by our group, where inhibition of 44% to 88% was observed [23,32]. Anyway, as previously mentioned, the model was efficient for the study of macrophage phenotypic reprogramming. Furthermore, we can suggest that the reprogramming or prevalence of the M1 phenotype of macrophages induced by CTX, observed in the present study, when administered concomitantly with tumor induction, explains the significant antitumor activity of the toxin in these models of solid and ascitic tumors, since macrophages are largely found in these tumors, which may represent up to 80% of the population of the tumor microenvironment (summarized in Figure 6).

Therefore, this toxin may be an important tool to modulate the phenotypic reprogramming of macrophages, which may favor the selection of new therapeutic targets for the correction of compromised immunity in different systems. The importance of combinations of different isoforms of both subunits (CA and CB) for the modulatory action of CTX on the phenotypic plasticity of macrophages is now being investigated.

## 4. Conclusions

This is the first scientific report about the effect of a single administration of CTX interfering with the phenotypic reprogramming of macrophages, favoring the M1 profile, which is important for the control of tumor progression. This evidence is supported by the inhibitory action of CTX on ascitic volume and the number of tumor cells and by the stimulatory effect on the production of cytokines crucial for the antitumor action of macrophages.

## 5. Materials and Methods

### 5.1. Crotoxin

CTX was purified from lyophilized *Crotalus durissus terrificus* snake venom, extracted from several adult specimens, supplied by the Laboratory of Herpetology, Butantan Institute and stored in a freezer at −20 °C. Purification of this toxin was performed according to the method described by Rangel-Santos et al. [38]. Briefly, an aliquot containing 10 mg of the venom was resuspended in 1 mL of Tris-HCl (50 mM, pH 7.0) and centrifuged at 10,000× *g* for 10 min (Ultra-Eppendorf Centrifuge) to remove insoluble material. The supernatant obtained was submitted to ion exchange chromatography on a 5 mL MONO-Q HR 5/5 column, in a FPLC system (Fast-performance liquid chromatography, Pharmacia), in 50 mM Tris-HCl buffer, pH 7.0, for CTX isolation. Proteins adsorbed to the resin were eluted by a linear gradient from 0 to 1 M NaCl and buffered with an equilibration buffer. Three main peaks were obtained (Peaks I, II and III), where peak II corresponds to the elution of CTX. Fractions of 1 mL per tube were collected, and the elution was monitored by reading the absorbance at 280 nm. Phospholipase A_2_ activity of CTX fractions was analyzed in in vitro assays using a synthetic chromogenic substrate. Tubes corresponding to CTX were pooled and dialyzed and their dose was determined by the Bradford method [39].

### 5.2. Animals

BALB/c mice were used (male, weighing between 25 and 30 g), provided by the Central Animal House of Butantan Institute and kept inside the cabins, with temperature control (22 ± 2 °C), ventilation and light (12 h of light and 12 dark hours), with free access to water and feed in the Laboratory of Pathophysiology for a period of at least two days before being used in the experiments. Mice were euthanized in a CO_2_ chamber (25%). The experimental protocols performed in this project were approved by the Ethics Committee of the Butantan Institute—CEUAIB (Protocol N^o^. 481108416).

### 5.3. Maintenance and Inoculation of Ehrlich Ascitic Tumor Cells

An aliquot of the Ehrlich Ascitic Tumor (EAT) cells was thawed, and its number was adjusted to 1 × 10^7^ cells in 0.5 mL of phosphate-buffered saline (PBS) and inoculated into the peritoneal cavity of mice to obtain an ascitic tumor. On the 13th day of evolution, the mice were euthanized, as described in Section 5.2. Ascitic fluid was collected from the peritoneal cavity (first ascites) and placed in a test tube containing 5% EDTA. The counting and determination of cell viability were performed in a Neubauer hemocytometer using Trypan Blue 0.1%, and the dose of tumor cells was adjusted to 1 × 10^7^ cells/0.5 mL in PBS and inoculated into the peritoneal cavity of the animals for the trials.

### 5.4. Temporal Evaluation of Ehrlich Ascitic Tumor

The EAT growth curve consisted of 3 periods: the lag phase (1–5 days after tumor cell inoculation), the log phase (6–10 days) and the terminal period (11–15 days), followed by the death of the host with a tumor [40]. Thus, the mice received the inoculation of EAT (1 × 10^7^/0.5 mL of PBS), and the temporal evaluation of the EAT was performed at different periods (3, 6 and 13 days after inoculation of tumor cells), as shown in the following Experimental Appendix A, represented in SM-2.

### 5.5. CTX Treatment

To evaluate the effect of the toxin on the reprogramming of macrophages obtained from Ehrlich’s tumor in the ascites form, the mice were inoculated with tumor cells (1 × 10^7^ tumor cells/0.5 mL of PBS) in the peritoneal cavity (tumor-bearing mice group). Then, the animals were immediately treated with CTX doses (0.9 µg/animal or 5.0 µg/animal, in 100 µL of saline, subcutaneously, on the upper back) according to the protocols below. Control groups (non-tumor-bearing mice) consisted of animals injected, i.p., (intraperitoneal) with a volume of 0.5 mL of sterile PBS in place of the tumor cells and treated with CTX (0.9 µg/animal or 5.0 µg/animal, in 100 µL of saline, subcutaneously, on the upper back) or the same volume of saline (control) under the same experimental conditions. The CTX doses used were based on previous studies [24,25,41]. The experimental groups were constituted as shown in the Experimental Appendix A, represented in SM-2.

#### Treatments CTX Performed at Different Periods

Evaluation of the effect of concomitant CTX treatment on the transition from the lag to log phase of development of Ehrlich tumor cell-induced ascites (1st treatment protocol): Mice were treated with CTX (0.9 or 5.0 µg/animal, in 100 µL of saline), concomitantly after inoculation of EAT (1st day), at a dose of 1 × 10^7^/0.5 mL of PBS, for the evaluation of different parameters on the 6th day of ascites development, as demonstrated in the Experimental Appendix A, represented in SM-2.

Evaluation of the effect of the concomitant treatment of CTX with the inoculation of Ehrlich tumor cells to obtain macrophages on the 13th day of EAT induction (2nd treatment protocol): To carry out the concomitant treatment protocol, the mice received a subcutaneous injection of CTX (0.9 or 5.0 µg/animal, in 100 µL of saline), immediately after inoculation of the EAT, at a dose of 1 × 10^7^/0.5 mL of PBS, as demonstrated in the Experimental Appendix A, represented in SM-2.

### 5.6. Obtaining Peritoneal Lavage from Non-Tumor-Bearing Animals

Healthy animals were euthanized in a CO_2_ chamber, and a small incision was made in the skin of the abdominal region and 5 mL of PBS was injected into the peritoneal cavity. After massaging the abdomen, the peritoneal lavage was collected using a polyethylene Pasteur pipette. The tubes were centrifuged at 250 × *g* for 10 min at 5 °C. The supernatant was discarded and the cell precipitate was resuspended and diluted 1:10 (*v*:*v*) with Trypan Blue (0.1%) and the total cell count was performed in a Neubauer hemocytometer with the aid of a light microscope.

### 5.7. Obtaining and Isolating Macrophages Obtained from Ascites Induced by Ehrlich’s Tumor

In animals with ascites, a small incision was made in the skin of the abdominal region to collect the ascitic fluid, which was collected with the aid of a polyethylene Pasteur pipette and placed in a test tube containing 5 mL of EDTA (ethylenediamine tetraacetic acid). The tubes were centrifuged at 250× *g* for 10 min at 5 °C and the supernatant was collected for later use. The cell pellet was resuspended in PBS or RPMI 1640 medium (Gibco, Billings, MT, USA), supplemented with 10% FBS (Fetal Bovine Serum, Cultilab, Campinas, Brazil), penicillin (100 U/mL) and streptomycin (100 μg/mL), and an aliquot of cells was diluted in a proportion of 1:80 (*v*:*v*) with 0.1% Trypan Blue, and the total and differential cell count (tumor and leukocytes) was performed in a Neubauer hemocytometer, with the aid of a light microscope. Samples containing more than 90% intact cells were used. Ascitic tumor macrophages were isolated by adherence in Petri dishes (140 × 15 mm) in the presence of RPMI culture medium, for 1 h at 37 °C, 5% CO_2_. After this period, the plate was washed 3 times with PBS to remove non-adherent cells. Adherent cells were removed gently with a spatula and ice-cold PBS. These were incubated with the different macrophage markers to identify the phenotypes of these cells.

### 5.8. Quantification of Nitric Oxide (NO^•^) Secreted in Ascites or Peritoneal Lavage

To determine the production of NO^•^, the protocol of Ding et al. [42] was used. After obtaining ascites or peritoneal lavage, 50 µL of samples were pipetted (in triplicate) and Griess’ reagent, in a 1:1 (*v*/*v*) ratio, was added. Then, the plate was read in an ELISA reader at 550 nm. The reading values were compared to a standard curve of sodium nitrite (NaNO_2_) and the results were expressed as µM of nitrite released in ascitic fluid and peritoneal lavage.

### 5.9. Enzyme Immunoassay (ELISA) for Cytokine Determination

This assay was performed for the determination of cytokines (IL-1β, IL-10 and TNF-α) in the supernatant of peritoneal lavage and ascites from animals treated with the toxin or not, according to Section 5.5. The determination of the dose of cytokines was performed by the ELISA, using reagent kits (R&D System, Minneapolis, MN, USA), following the specific recommendations of the manufacturer.

### 5.10. Macrophage Immunophenotyping

Cells were isolated as described in Section 5.6 and Section 5.7 and analyzed by flow cytometry to characterize the phenotypes. For that, the suspensions were centrifuged and resuspended in RPMI 1460 medium, 1 × 10^6^ cells were plated in a 96-well plate and incubated with anti-FcγRII/III antibody for 30 min at 4 °C. After this period, the plate was centrifuged (470× *g*, 5 min, 10 °C) and the samples were resuspended in 20 µL of RPMI 1640 medium and incubated with 0.25 µL/well of anti-CD45 antibody (leukocytes) and 0.5 µL/well of anti-F4/80 antibodies (macrophages), anti-CD68 (M1-macrophages) and anti-CD206 (M2/TAM-macrophages) for 30 min, at 4 °C, in the dark. Then, the cells were washed once with 100 µL PBS-1% BSA, centrifuged (470× *g*, 5 min, 10 °C) and resuspended in 200 µL PBS containing 1% paraformaldehyde. Samples were read and analyzed by flow cytometry (BD Accuri™ C6, Ann Arbor, MI, USA) using the cFlow software version 3.0 (BD). Fluorochrome compensation was performed with peritoneal lavage cell populations, with unique labels for each fluorochrome. In all experiments, 5000–10,000 events were acquired from each sample and the data were analyzed using FlowJo software version 10.6. The M1 and M2 macrophage populations in different experimental groups were defined according to the following strategy of analysis: Firstly, a cell population was gated using forward scatter (FSC) versus side scatter (SSC) parameters followed by the selection of leukocytes using CD45^+^ marker versus SSC-A parameters. Then, the gated CD45^+^ cells were used to analyze the double expression of F4/80^+^CD68^+^ molecules for M1 macrophages and F4/80^+^CD206^+^ for M2 macrophages.

### 5.11. Statistical Analysis

Statistical analyses were performed using InStat software version 3.0 and GraphPad Prism software version 8.4.0. For multiple comparisons, one-way analysis of variance (one-way ANOVA) was used, followed by the Bonferroni test. For comparisons between the two groups, the unpaired Student’s *t* test was used. Results were expressed as mean ± standard error of the mean (SEM). Differences were considered statistically significant with *p* < 0.05.

## Figures and Tables

**Figure 1 toxins-15-00616-f001:**
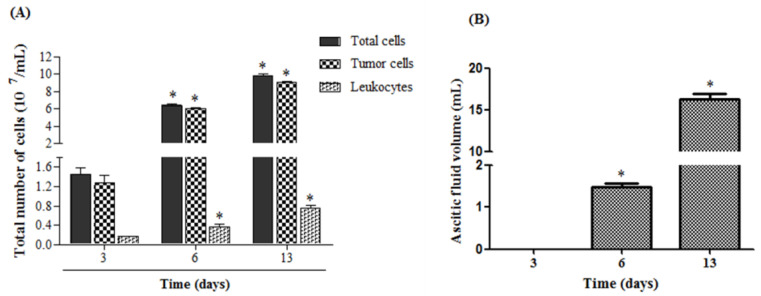
Temporal evolution of ascites induced by Ehrlich tumor cells. The animals were inoculated with EAT (1 × 10^7^ cells/0.5 mL), and after the 3rd, 6th and 13th day of tumor cell inoculation, it was determined in (**A**) the total and differential count (leukocyte and tumor) in a hemocytometer Neubauer and in (**B**) the volume of ascitic fluid collected from the abdominal cavity. Values are expressed as mean ± SEM of 3–4 animals per group. * *p* < 0.001, compared to the respective groups over the 3-day period.

**Figure 2 toxins-15-00616-f002:**
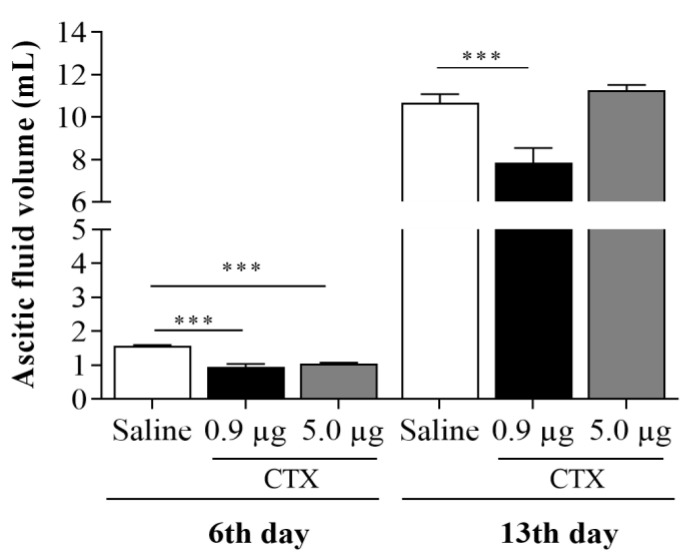
Effect of CTX on the volume of the ascites fluid. The animals were inoculated with EAT (1 × 10^7^ cells/0.5 mL) and treated concomitantly with different doses of CTX (0.9 µg/animal and 5.0 µg/animal, in 100 µL of saline, s.c.) or saline (100 µL). On the 6th and 13th day of tumor cell inoculation, ascitic fluid was collected from the peritoneal cavity and the total volume was measured. *** *p* < 0.001 compared to the control group (EAT + saline) and the CTX-treated group. Values are expressed as mean ± SEM of 5–8 animals per group.

**Figure 3 toxins-15-00616-f003:**
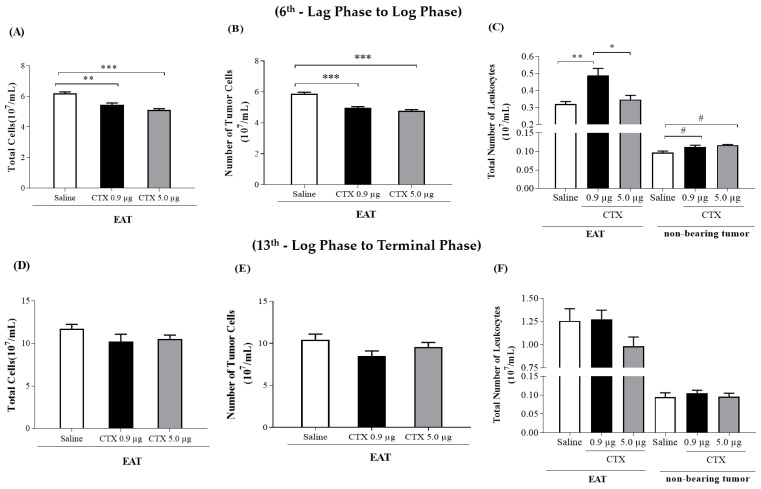
Effect of CTX on the number of cells obtained from EAT in mice. The animals were inoculated with EAT (1 × 10^7^ cells/0.5 mL PBS) or injected, i.p., with PBS and treated concomitantly with different doses of CTX (0.9 µg/animal and 5.0 µg/animal in 100 µL of saline, s.c.) or saline (100 µL). On the 6th or 13th day of the EAT inoculation, the total counts (**A**,**D**) of tumor cells (**B**,**E**) and leukocytes (**C**,**F**) were determined from the ascitic fluid and the peritoneal cavity (non-carrier animals) in a Neubauer hemocytometer. Values are expressed as mean ± SEM of 3–9 animals per group. * *p* < 0.05 compared to the CTX-treated group. ** *p* < 0.01 compared to the control group (EAT + saline). *** *p* < 0.001 compared to the control group (EAT + saline) and the CTX-treated group. # *p* < 0.05 compared to the control group (PBS + saline).

**Figure 4 toxins-15-00616-f004:**
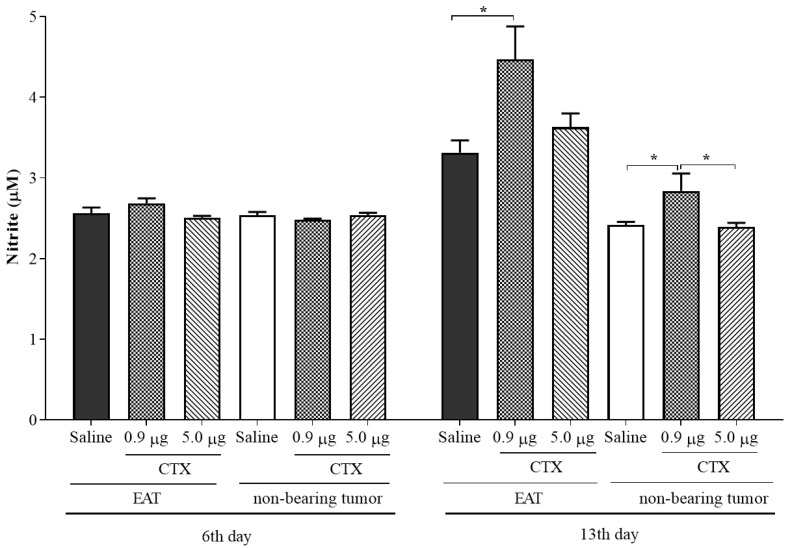
Effect of CTX on nitric oxide production. The animals were inoculated with EAT (1 × 10^7^ cells/0.5 mL PBS) or injected, i.p., with PBS (non-tumor-bearing animals) and treated concomitantly with different doses of CTX (0.9 µg/animal and 5.0 µg/animal in 100 µL saline, s.c.) or saline (100 µL). After the 6th or 13th day of tumor cell inoculation, the supernatant from each group was transferred to a reading plate and Griess reagent (1:1, *v*/*v*) was added. Then, the plate was read in an ELISA reader at 550 nm. The reading values were compared with a standard curve of sodium nitrite (NaNO_2_) and the results were expressed as µmoles of nitrite in the ascitic fluid (tumor-bearing animals) or in the peritoneal lavage of the non-tumor-bearing animals. Values are expressed as mean ± SEM of 4–9 animals per group. * *p* < 0.05, compared to the respective control (EAT + saline and PBS + saline) and the experimental groups treated with CTX.

**Figure 5 toxins-15-00616-f005:**
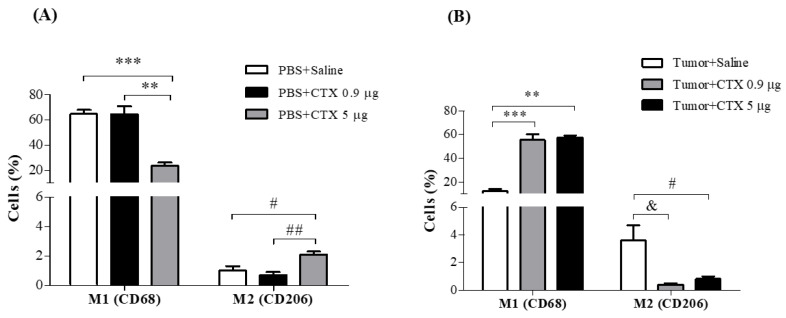
Effect of CTX on macrophage reprogramming. (**A**) Animals injected intraperitoneally with PBS (0.5 mL) (non-tumor-bearing animal group) and (**B**) tumor-bearing animals group (inoculated with EAT 1 × 10^7^ cells/0.5 mL); both were treated concomitantly with different doses of CTX (0.9 and 5.0 µg/animal in 100 µL of saline, s.c.) or saline (100 µL). On the 13th day of inoculation with PBS or tumor cells and concomitant treatment with CTX, the adherent cells from the intraperitoneal cavity were incubated with different mAbs for the macrophage phenotypes (anti-CD45, anti-F4/80, anti-CD68 and anti-CD206) labeled with distinct fluorophores and analyzed by flow cytometry. Values are expressed as the percentage of positive cells ± SEM of 4–6 animals per group in the figures of M1 and M2 phenotypes. ** *p* < 0.05, compared to the group treated with the dose of CTX 0.9 µg/animal. ^#^ *p* < 0.05, compared to the control group (saline). ^&^ *p* < 0.01, compared to the control group (saline). ^##^ *p* < 0.05, compared to the group treated with CTX. *** *p* < 0.001, compared to the control group (saline).

**Figure 6 toxins-15-00616-f006:**
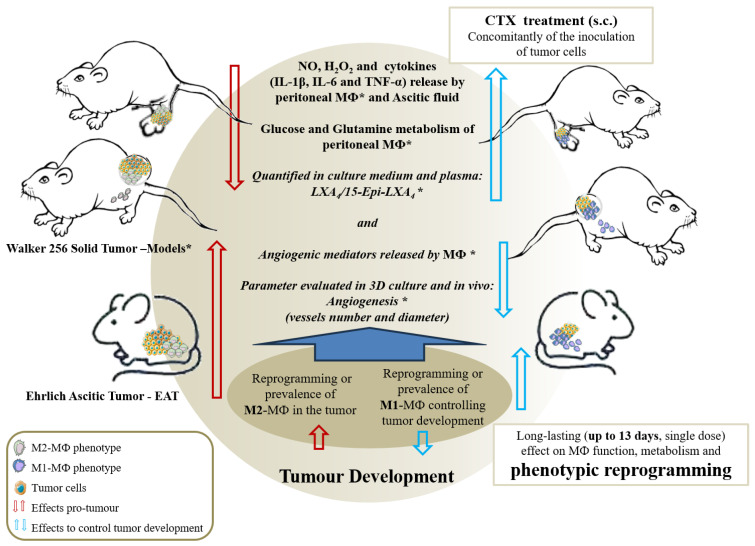
Scheme proposed for the importance of phenotypic reprogramming of macrophages induced by CTX in the tumor microenvironment. The data found in the present study demonstrate that CTX induces phenotypic reprogramming of macrophages, with a prevalence of M1 macrophages in the tumor microenvironment. This prevalent M1 profile agrees with the CTX-induced modifications on the metabolism and secretory activity of peritoneal macrophages obtained from tumor-bearing animals * [23,32]. The stimulatory action on the metabolism of macrophages was characterized by an increase in the release of H_2_O_2_, the production of NO, the secretion of pro-inflammatory cytokines (IL-1β, TNF-α and IL-6) and the increased maximum activity of hexokinase, glucose-6-phosphate dehydrogenase and citrate synthase, accompanied by an increased secretion of LXA_4_ and 15-Epi-LXA_4_ by these cells, also characterized in in vitro studies [30,36]. All these mediators lead to the inhibition of tumor development. The present study shows for the first time that CTX induces phenotypic reprogramming, explaining the metabolism and secretory activity of macrophages compatible with the M1 profile, as previously demonstrated, with long-lasting action, since they are observed up to 13 days after the administration of a single dose of toxin [37].

**Table 1 toxins-15-00616-t001:** Effect of CTX on cytokine secretion from the peritoneal cavity of non-tumor-bearing animals and on the ascitic fluid of tumor-bearing animals after 6th day of inoculation of EAT cells.

6th Day after EAT Inoculation						
	Non-Tumor-Bearing Mice			Tumor-Bearing Mice		
Cytokines (pg/mL)	Saline	CTX 0.9 µg/Animal	CTX 5.0 µg/Animal	Saline	CTX 0.9 µg/Animal	CTX 5.0 µg/Animal
IL-1β	14.5 ± 1.50	15.8 ± 3.04	14.0 ± 3.24	7.3 ± 1.65	8.0 ± 2.04	6.5 ± 0.87
IL-10	13.1 ± 10.7	9.2 ± 9.2	30.6 ± 15.1	84 ± 29.8	N.D. *	N.D. *
TNF-α	252.8 ± 37	247.5 ± 57.2	297.3 ± 22	114.5 ± 30.4	65.7 ± 4.8	50.1 ± 8.2

Animals were injected, i.p., with PBS (non-tumor-bearing animals) or inoculated (tumor-bearing animals) with EAT (1 × 10^7^ cells/0.5 mL PBS) and treated concomitantly with different doses of CTX (0.9 µg/animal and 5.0 µg/animal in 100 µL saline, s.c.) or saline (100 µL-control). After the 6th day of PBS or EAT inoculation, peritoneal lavage or ascitic fluid, respectively, from each group was collected to determine the dose of cytokines (IL-1β; IL-10 and TNF-α) by ELISA. Values are expressed as mean ± SEM of 4 animals per group. * *p* < 0.05, compared to the control group (PBS + saline; EAT + saline).

**Table 2 toxins-15-00616-t002:** Effect of CTX on the cytokine secretion from the peritoneal cavity of non-tumor-bearing animals and on the ascitic fluid of tumor-bearing animals after the 13th day of inoculation of EAT cells.

13th Day after EAT Inoculation						
	Non-Tumor-Bearing Mice			Tumor-Bearing Mice		
Cytokines (pg/mL)	Saline	CTX 0.9 µg/Animal	CTX 5.0 µg/Animal	Saline	CTX 0.9 µg/Animal	CTX 5.0 µg/Animal
IL-1β	13.8 ± 3.8	13.3 ± 3.7	13.7 ± 0.24	5.8 ± 1.44	3.5 ± 1.85	2.5 ± 0.50
IL-10	69.8 ± 68.8	42.6 ± 42.6	1.9 ± 1.9	342 ± 75	309 ± 61	267 ± 54
TNF-α	298 ± 99	167 ± 44	215 ± 7.3	14.3 ± 3	18 ± 5	24 ± 1.3

Animals were injected, i.p., with PBS (non-tumor-bearing animals) or inoculated (tumor-bearing animals) with EAT (1 × 10^7^ cells/0.5 mL PBS) and treated concomitantly with different doses of CTX (0.9 µg/animal and 5.0 µg/animal in 100 µL saline, s.c.) or saline (100 µL-control). After the 13th day of PBS or EAT inoculation, peritoneal lavage or ascitic fluid, respectively, from each group was collected to determine the dose of cytokines (IL-1β; IL-10 and TNF-α) by ELISA. Values are expressed as mean ± SEM of 4 animals per group.

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
