# Peer review of "Crotoxin Modulates Macrophage Phenotypic Reprogramming"

_toxins, 2023, doi:10.3390/toxins15100616_

Round 1
Reviewer 1 Report
In this manuscript authors have explored the impact of a promising antitumor and abundant toxin isolated from Crotalus durissus terrificus snake venom on the macrophage phenotypic reprogramming. There are several interesting publications addressing its pharmacological properties and action on cancer cells and cells that participate in the innate and adaptive response. I have read with interest this paper, which broadens our understanding of the biomedical applications of a snake venom protein, with particular emphasis on relevant events associated with tumour progression, such as the phenotypic reprogramming of macrophages. In summary, the research is important and novel, but authors should clarify diverse key points before its publication, mainly those related to experimental design.
1. Abstract should focus more in results and the implications of the main findings. In the current version of this manuscript, half of the abstract section describes previous knowledge. The novelty of this study should be highlighted in this section.
2. Line 13. Please revise the meaning of concentration and dose. The unit used are not appropriated to describe concentrations.
3. Line 13. I do not think it is necessary to include the word different. The authors detailed both doses, which are clearly different.
4. Line 14. at the first day. Please specify the timepoint (24 h?).
5. Abstract section. Authors should add some perspectives or future directions based on their conclusions in the abstract section. This work probably opens new research avenues or applications.
6. Key contribution sections. Basically, the authors have described their results. This must be rewritten. What are the contributions or main implications of their findins?
7. Lines 36-39. This sentence is too long and hard to understand. For example, "perform activities that can prevent the establishment and spread of tumor" (what are these activities?). In the same direction, "can stimulate functions that favor tumor growth" (which functions").
8. line 58. 2 must be a subscript.
9. line 59. Scientific name are always italicized .
10. Line 65. I am interested in understanding the long-term immunomodulatory effect of this crotoxin. I tried to check the reference 23 out, but there is no title or relevant information to get access. How did the authors classify this activity as a long-term effect?
11. Line 67-70. Please avoid one-sentence paragraph. On the other hand, this sentence is too long and hard to understand.
12. line 70. Please clarify long period.
13. The figures are presented in low-quality. Please provide high-quality images.
14. Please specify the statistical test used to compare the data in the figure legend.
15. Line 151. Concentrations? There is no unit of volumen. Dose?
16. Table 1 and 2. Please details the unit of cytokine levels.
17. How do the authors explain the values of 0 for IL-10? Please discuss this.
18. Authors should standarise the presentation of figures and tables. For example, the used different two ways to express the dose. Please revise this point.
19. Lines 261-262. This is not clear for me. What are the results associated with phenotypic profiles and what are the findings associated with the funcitonality of macrophages?
20. Lines 263- 273. Authors have raised a really important point that serve as a basis for understanding their findings and the motivation behind this study. I was really curious to understand how the authors choose theses concentrations. In this sense, I believe the rationale behind this study must be presented in the introduction section. The state-of-the art is very important to contextualise and must be included before the results section. Definitely, this paragraph was crucial for my understanding. However, I am still curious about the experimental design. There are different manuscripts published describing the action of crotoxin. In the context of this manuscript, the manuscript by de Araujo Pimenta et. al is very important (please see: de Araújo Pimenta, L., de Almeida, M.E.S., Bretones, M.L. et al. Crotoxin promotes macrophage reprogramming towards an antiangiogenic phenotype. Sci Rep 9, 4281 (2019). https://doi.org/10.1038/s41598-019-40903-0). Why did the authors not take into account the concentrations used in this research? Comparisons and discussions about this are enriching for the field.
21. Discussion sections is too long. Please revise carefully this section. Some words are repeated several times, for instance. Regarding and so on.
22. Discussion section. In view of objective of this research (to expand the mechanisms involved in the antitumor effects of CTX) described in lines (67-70), the authors should include a summary figure that illustrates the current knowledge of the mechanism of action of CTX and also highlight their contributions based on the results of this research. Therefore, this new figure will help readers to have an integrative view of the mechanisms involved in the antitumor effect and facilitate the adequate assessment of the importance of the results described in this manuscript.
23. Conclusions. Authors must emphasise the implications of their findings.
24. Line 399. described by... (include author's name)
25. Lines 409. What was the synthetic substrate employed?
26. Lines 410. by the method of ... Please include author's name.
27. Authors mentioned that there are different isoforms of CTXs. In this sense, the action, affinity, specificity and activities of each toxin isoform can be variable. What was the isoform used in this study?
28. Did the authors assess the purity level of toxin? How did they do that?
29. Line 422. Why did the authors choose this day? I am venom researcher, and I curious about that. Is it because of the toxin-mediated mortality?
30. Based on the description of the EAT growth curve provided by the authors, I do not understand the selection of the timepoints (3, 6 and 13). Why not 5, 10 and 14 or 6, 11 and 14?
31. The experimental schemes can be presented as supplementary material. Also, authors needs to improve their quality. Please check the red colour. What is the purpose of highlight these words?
32. Line 491. Please include author`s name
33. line 515. This is a wide-range. How was the number of events defined? Did the authors used the same number of events to compare the groups?
34. 5.11. Please clarify when the Tukey or Bonferroni test were used.
Author Response
"Please see the attachment."

Reviewer 2 Report
The submitted manuscript examines the effects of Crotoxin on in vivo macrophages in BALB/C mice plus or minus challenge with syngeneic Ehrlich Ascitic Tumors (EAT). Observed difference in macrophage polarization and tumor growth dependent on toxin concentration
Comments:
1) Text states “it was observed that only the concentration of 0.9 μg/animal of CTX was able to significantly reduce (27%) the volume of ascites, when compared to the control group.” This does not match figure 2 where it is not denoted that there is a significant difference between control and 0.9ug/animals. Instead figure denotes a significant difference between volumes obtained from 0.9ug/animals and 5ug/animals.
2) In section 2.4 state that NO is coming from macrophages. This is the likely source of NO but since it is being measure from the ascites or peritoneal fluid you don’t know for sure what the cell type of origin is. Text should be changed to not definitively state that this is a change in macrophage NO production.
3) Abstract should be rewritten. It is hard to follow in current state. Most of the information in parenthesis could probably be depleted—the granularity is not needed in the abstract. However be more specific about the modulation of M1 and M2 phenotype instead of saying these phenotypes were modulated. Lastly in the abstract it is stated “This study show that CTX interferes with the 18 phenotypic reprogramming of macrophages, as well as with the secretory state of these cells…”. Similar to point 2 there is no evidence presented that the changes in cytokine levels is due to changes in production by macrophages.
4) Lastly it would be interesting to determine levels of CD8+ T cells and FoxP3+ Tregs which could be impacted by macrophage polarization and then impact tumor growth.
Quality of English fine, but the abstract could be improved to better summarize the manuscript.
Author Response
"Please see the attachment."

Reviewer 3 Report
I have reviewed the manuscript “Crotoxin modulates macrophage phenotypic reprogramming” and have several concerns:
Abstract
Line 4: “Macrophage plasticity is a fundamental feature for the immune defense”. it is better to write: “Macrophage plasticity is a fundamental feature for the immune response”.
Line 8: “control of tumor progression” line 12: “reprogramming of macrophages in the tumor microenvironment” Line 19: …”events involved with tumor progression”. The type of tumor must be noted. Mesenchymal? epithelial?. When trying to investigate the responses found in a form of sarcoma, it is better to specify that it is a mesenchymal tumor.
Keywords:
Line 21: Crotoxin. Macrophages. Reprogramming: They are already in the title of the manuscript. It is better to use other words, to expand the possibilities of being viewed by a reader.
Discussion
The immune system must be very competent to kill cancer cells. Even cancer immunotherapy is based on activating the immune system (check-points therapy). Why inhibiting an M1 phenotype of macrophages, which is proinflammatory, can help slow down the growth of a neoplasm? this is hard to conclude. This should be clarified in the manuscript.
Lines 18-20: “This study show that CTX interferes with the phenotypic reprogramming of macrophages, as well as with the secretory state of these cells, influencing events involved with tumor progression”. Lines 390-392: …”effect of a single administration of CTX interferes with the phenotypic reprogramming of macrophages, as well as with the cellularity present in the ascitic fluid and the secretory state of these cells, influencing the events involved with tumor progression”
The most important conclusion of the article is the one presented in the title: Crotoxin modulates macrophage phenotypic reprogramming…
Author Response
"Please see the attachment."

Reviewer 4 Report
The authors have studied the antitumor effects of Crotoxin (CTX). In this study, we are investigating its antitumor activity and effects on macrophage phenotypes using an EAT tumor-bearing mouse model. It is an easy-to-understand paper, but I think that the reader's understanding can be better understood by responding to the following comments.
1. L102-107 are duplicates of L91-96 and should be deleted.
2. According to the author, there are 16 types of CTX, but I would like you to explain in the introduction how the CTX used in this experiment was selected.
3. Please explain why CTX is administered subcutaneously while EAT is administered intraperitoneally.
4. It is natural that research has various limitations, so I would like you to add the limitations of this research result as a paragraph in the discussion.
Author Response
"Please see the attachment."

Round 2
Reviewer 1 Report
I have no additional comments. Authors have carefully addressed all points I raised on the previous report. Overall, the manuscript has considerably improved. The figure summarising the state-of-the-art, findings and novel contributions helps readers to understand the objective and conclusions of this interest study.
Reviewer 3 Report
The new version of the manuscript is adequate. The authors respond appropriately to the suggestions.